# Temporal patterns, spatial risks, and characteristics of tegumentary leishmaniasis in Brazil in the first twenty years of the 21st Century

**Vinícius Silva Belo**[1]*, **Fábio Raphael Pascoti Bruhn**[2], **David Soeiro Barbosa**[3], **Daniel Cardoso Portela Câmara**[4], **Taynãna César Simões**[5], **Lia Puppim Buzanovsky**[6,7], **Anna Gabryela Sousa Duarte**[1], **Saulo Nascimento de Melo**[1], **Diogo Tavares Cardoso**[3], **Lucas Edel Donato**[8], **Ana Nilce Silveira Maia-Elkhoury**[7], **Guilherme Loureiro Werneck**[9,10]

**1** Campus Centro-Oeste Dona Lindu, Universidade Federal de São João del-Rei, Divinópolis, Minas Gerais, Brazil, **2** Departamento de Veterinária Preventiva, Faculdade de Veterinária, Universidade Federal de Pelotas, Pelotas, Rio Grande do Sul, Brazil, **3** Departamento de Parasitologia, Instituto de Ciências Biológicas, Universidade Federal de Minas Gerais, Belo Horizonte, Minas Gerais, Brazil, **4** Laboratório de Imunologia Viral, Instituto Oswaldo Cruz, Fundação Oswaldo Cruz, Rio de Janeiro, Rio de Janeiro, Rio de Janeiro, Brazil, **5** Instituto René Rachou, Fundação Oswaldo Cruz, Belo Horizonte, Minas Gerais, Brazil, **6** Centro Pan-Americano de Febre Aftosa, *Organização Pan-Americana da Saúde*, Rio de Janeiro, Rio de Janeiro, Brazil, **7** *Organização Pan-Americana da Saúde*, Rio de Janeiro, Rio de Janeiro, Brazil, **8** Ministério da Saúde, Brasília, Federal District, Brazil, **9** Departamento de Epidemiologia, Instituto de Medicina Social, Universidade do Estado do Rio de Janeiro, Rio de Janeiro, Rio de Janeiro, Brazil, **10** Instituto de Estudos em Saúde Coletiva, Universidade Federal do Rio de Janeiro, Rio de Janeiro, Rio de Janeiro, Brazil

* viniciusbelo@ufsj.edu.br

## Abstract

### Background

Tegumentary leishmaniasis (TL) is a significant public health issue in Brazil. The present ecological study describes the clinical and epidemiological characteristics of TL cases reported in the country, and analyzes the spatial and temporal patterns of the incidences and risks of occurrence across the five geopolitical regions and 27 federative units.

### Methodology/Principal findings

Data regarding new cases of TL notified between 2001 and 2020 were obtained from the Information System for Notifiable Diseases of the Brazilian Ministry of Health. Joinpoint and spatial and temporal generalized additive models were used to establish trends in the evolution of TL during the target period. The incidence rate for the entire period was 226.41 cases/100,000 inhabitants. All regions of Brazil showed trends of decreasing incidence rates, albeit with fluctuations at specific times, with the exception of the Southeast where rates have increased since 2014, most particularly in Minas Gerais state. The disease was concentrated predominantly in the North region, with Acre state leading the incidence rank in the whole country, followed by Mato Grosso (Midwest), Maranhão and Bahia (Northeast) states. The spatial distribution of the risk of TL occurrence in relation to the annual averages was relatively stable throughout the period. The cutaneous form of TL was predominant and

**Data Availability Statement:** The data are available from the Information System for Notifiable Diseases of the Brazilian Ministry of Health

(SINAN)( - http://www.portalsinan.saude.gov.br -; from the Mortality Information System from the Brazilian Ministry of Health (SIM) - http://tabnet. datasus.gov.br/cgi/deftohtm.exe?sim/cnv/obt10uf. def - and from the Brazilian Institute of Geography and Statistics (IBGE) - https://sidra.ibge.gov.br/.

**Funding:** This work was funded by Organização Pan- Americana da Saúde (SCON2021-00153 to VSB)(https://www.paho.org/pt/brasil). This work was financed the Coordenação de Aperfeiçoamento de Pessoal de Nível Superior – Brasil (CAPES) – Code 001. The funder(s) had no role in study design, data collection and analysis, decision to publish, or preparation of the manuscript.

**Competing interests:** The authors have declared that no competing interests exist.

cases most frequently occurred in rural areas and among men of working age. The ages of individuals contracting TL tended to increase during the time series. Finally, the proportion of confirmations by laboratory tests was lower in the Northeast.

## Conclusion/Significance

TL shows a declining trend in Brazil, but its widespread occurrence and the presence of areas with increasing incidence rates demonstrate the persistent relevance of this disease and the need for constant monitoring. Our findings reinforce the importance of temporal and spatial tools in epidemiologic surveillance routines and are valuable for targeting preventive and control actions.

## Author summary

Leishmaniasis, a neglected tropical disease caused by parasitic protozoa, exists in various clinical forms, the most common of which is tegumentary leishmaniasis (TL). The lethality of TL is low, but the skin lesions may produce physical deformities and stigma. Brazil has the highest number of TL cases in the American continent and the disease is present in all federative units. However, studies performed so far have been restricted to localized areas of Brazil and do not provide a broad view of the situation in the country. Therefore, we described the clinical and epidemiological characteristics of TL cases in Brazil and evaluated the spatial and temporal patterns from 2001 to 2020. Our findings show that the occurrence of TL declined between 2001 and 2020 in all regions of the country except for the Southeast where incidence rates have recently increased. We have identified those areas of the country where the population remains at greater risk of contracting the disease and have found that the situation has changed little over the years. The knowledge gained about the spatial and temporal variations of the incidence of TL, the profiles of those affected by the disease, and the diagnostics and treatment employed might be valuable to improve the policies and actions against the disease.

## Introduction

Tegumentary leishmaniasis (TL), a neglected tropical disease that is associated with poverty and poor living conditions, is considered one of the six most important infectious diseases in the world [1]. It is estimated that 1 million cases of TL occur annually [2] generating a significant socioeconomic burden on the affected populations [1, 3, 4]. Despite being endemic in 90 countries worldwide, more than 85% of TL cases occur in just 10 countries, one of which is Brazil where the disease is considered a significant public health problem and requires compulsory notification [5, 6]. In addition, the disease is of concern in the country because it has high incidence rates in large parts of the territory, occurs in all federative units (states and the Federal District), permeates all age groups and has multiple transmission patterns [5, 7].

The etiological agents of TL are protozoa of the genus *Leishmania* that are transmitted to humans and other hosts by insects of the family Psychodidae. In Brazil, the main protozoa are *Leishmania (Viannia) braziliensis* (present in all transmission areas), *L. (V.) guyanensis* (restricted to the Amazon basin) and *L. (Leishmania) amazonensis* (limited to specific states in some regions of the country) [7, 8]. The clinical forms of TL are described as cutaneous

(localized, disseminated or diffuse) and mucosal or mucocutaneous [5], while the development of the disease depends on the *Leishmania* species, the parasitic load and factors relating to the vector and hosts [9, 10].

The conventional therapy for cutaneous and mucocutaneous leishmaniasis in Brazil is meglumine antimoniate (MA) [7], a drug that is difficult to administer and, being highly toxic, can induce severe adverse effects [11–13]. Treatment with intravenous liposomal amphotericin B, which is less toxic than MA is recommended for individuals over 50 years of age and those suffering from specific clinical conditions [7], but it is rather expensive [14]. Recently, the alkylphosphocholine (alkyl-PCs) miltefosine has been indicated for the treatment of cutaneous leishmaniasis in Brazil and can be administered orally, but the drug presents some disadvantages including teratogenic effects, high-cost and potential protozoa resistance [15–17].

Considering the incidence of TL in Brazil and the challenges in treatment options, it is essential that surveillance and prevention actions be constantly improved. Implementation of effective protective measures demands knowledge of the characteristics of notifications and the epidemiology of the disease, particularly regarding the spatial and temporal patterns of new cases, which better indicate the transmission patterns [18, 19]. However, literature concerning the epidemiology of TL in Brazil is based predominantly on data from local surveys [19, 20] and there are no studies that evaluated the spatial and temporal patterns of TL in the country over 20 years at the subnational level.

In light of the above, we aimed to describe the clinical and epidemiological characteristics of TL cases notified in Brazil and evaluate the spatial risks and temporal patterns from 2001 to 2020.

## Methods

### Ethical considerations

Ethical approval for this ecological study was not required because the research involved data derived from secondary sources that are freely available to the public.

### Study area description and data sources

Brazil is the sixth most populous country in the world with more than 215 million inhabitants (Instituto Brasileiro de Geografia e Estatistica (IBGE), 2022). The territory is divided into five major geopolitical regions (South, Southeast, Midwest, Northeast and North), comprising 26 states and the Federal District (Fig 1), and 5,570 municipalities. Population data were obtained from the IBGE (https://www.ibge.gov.br/), while epidemiological data regarding the TL cases notified during the period 2001 to 2020 were extracted from the Information System for Notifiable Diseases (SINAN; http://www.portalsinan.saude.gov.br). For the analysis of the number of deaths and mortality rates, data was extracted from the Mortality Information System (SIM; http://tabnet.datasus.gov.br/cgi/deftohtm.exe?sim/cnv/obt10uf.def).

The original databases were assembled into a single standardized database that included only cases classified as new and from which duplicate and inconsistent records were removed.

### Clinical and demographic characteristics of the cases

Descriptive statistics were used to explore the distribution of the incidence rates of TL within Brazil and by demographic, clinical and epidemiological characteristics. The numbers of cases and deaths, incidence rates (total number of cases/mid-period population x 100,000), mortality rates (number of deaths/mid-period population x 100,000), annual mean number of cases (AMC), average annual incidence rate (AMC/ mid-period population x 100,000) and case

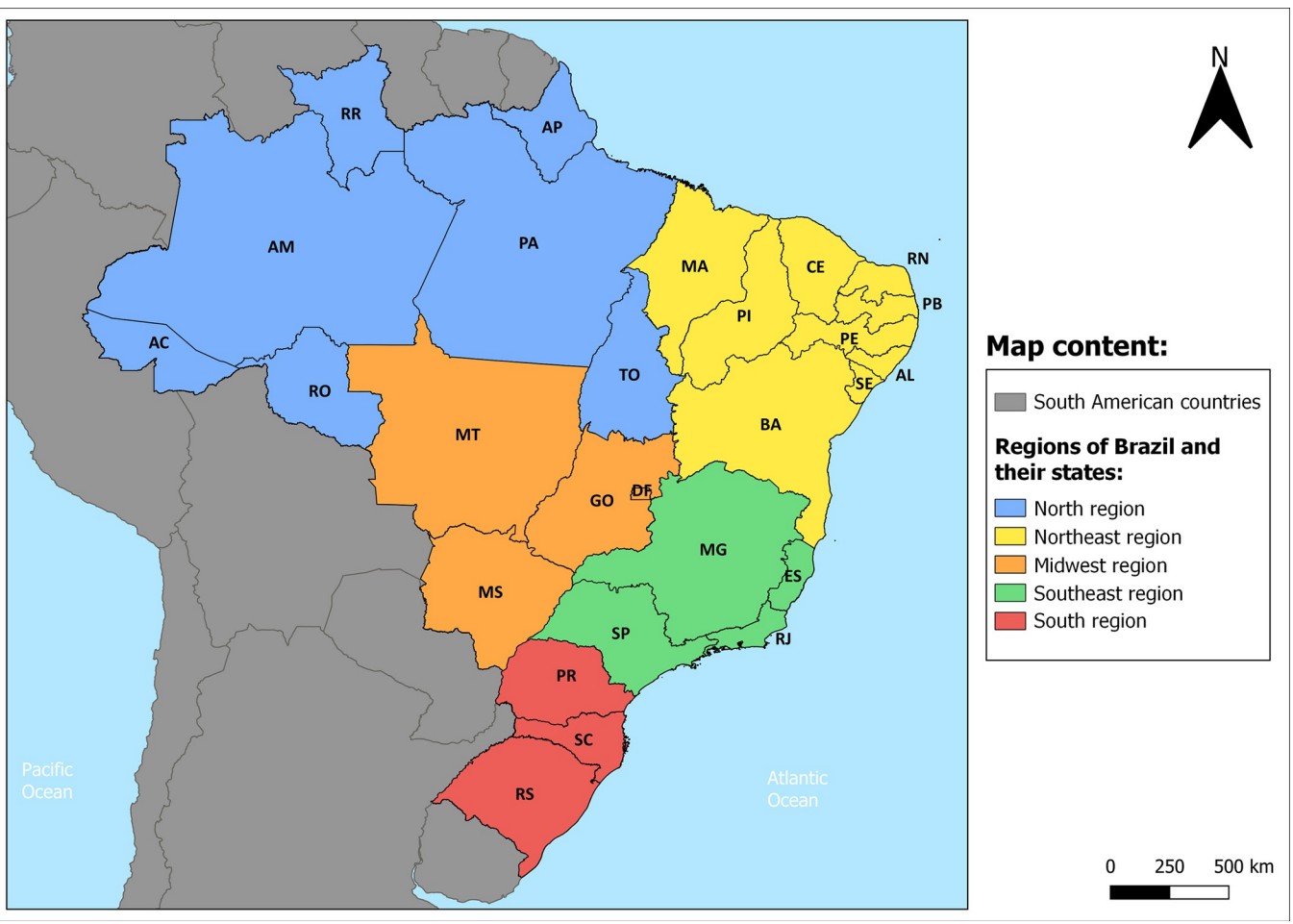

**Fig 1. Geopolitical regions and federative units of Brazil.** The map was built using the free and open source QGIS software (https://www.qgis.org/en/site/) based on shapefiles obtained from Instituto Brasileiro de Geografia e Estatística -IBGE- (https://www.ibge.gov.br/geociencias/organizacao-do-territorio/malhas-territoriais/15774-malhas.html) and from the Agência Nacional de Águas e Saneamento Básico (ANA, Ministry of Regional Development—Brazil) (https://metadados.snirh.gov.br/geonetwork/srv/api/records/7cfd53c4-b4e1-4aba-a79b-857a19649df6). Abbreviations: AC, Acre; AL, Alagoas; AM, Amazonas; AP, Amapá; BA, Bahia; CE, Ceará; DF, Distrito Federal; ES, Espírito Santo; GO, Goiás; MA, Maranhão; MG, Minas Gerais; MS, Mato Grosso do Sul; MT, Mato Grosso; PA, Pará; PB, Paraíba; PE, Pernambuco; PI, Piauí; PR, Paraná; RJ, Rio de Janeiro; RN, Rio Grande do Norte; RO, Rondônia; RR, Roraima; RS, Rio Grande do Sul; SC, Santa Catarina; SE, Sergipe; SP, São Paulo; TO, Tocantins.

lethality rates (number of deaths/total cases x 100) during the entire study period were determined for the whole country, and its regions and federative units. The standardized annual incidence rates of TL in the regions were constructed, according to the direct method, taking into account the age structure of the Brazilian population as published in the 2010 census. Choropleth maps depicting the annual incidence rates of TL in the federative units using Jenks natural breaks algorithm were constructed using the QGIS software, version 3.1.0 (https://www.qgis.org/en/site/).

Notified cases of the disease in Brazil were initially stratified by five-year period (2001–2005, 2006–2010, 2011–2015 and 2016–2020) and region. Then, data was described according to the age of the patient using the mean, median, standard deviation and interquartile range. Categorical variables (sex, ethnicity/skin color, location, education, disease confirmation, co-infection, treatment, clinical forms and evolution) were described in terms of absolute numbers and percentages.

## Time trends

The temporal trend of TL in Brazil and its regions and federative units during the study period were established using Joinpoint models and generalized additive models (GAM). Joinpoint regression combines different components of linear terms of the time-series. The inflection points (joinpoints) reveal moments in time when the trend change [21, 22]. Log-linear Join-point plots of the annual age-standardized incidence rates of TL were constructed and the best-fit models were selected on the basis of Monte Carlo permutation tests with 4999 iterations [23]. Increasing (lines with positive slope), decreasing (lines with negative slope) or stable (lines with null slope) patterns of the occurrence of TL during the target period were evaluated using annual percentage change (APC) measures to summarize trends in incidence rates of the disease at different time periods. In cases where there were at least one statistically different inflection points (p < 0.05), average annual percentage change (AAPC) measures were also determined for the entire series [24]. The 95% confidence intervals (95%CIs) were obtained for both APC and AAPC measures, and statistical significance was defined in cases where the 95%CIs did not include the null value.

With regard to GAM analysis, the smoothing curves of the expected effect of the year of notification on the incidence rate defined the temporal graphic trajectory [25]. In the temporal GAM models, the response variables were observed cases and expected cases after standardization according to notification year. Based on Akaike Information Criterion (AIC) values, negative binomial distribution models showed better fit in comparison with Poisson distribution models. The linear predictor variable of the models was the year of notification with a smoothing function (spline), whereas the offset term was the natural logarithm of the resident population in each year. The resulting curves showed the estimated risk of occurrence of TL in each year compared with the risk over the entire period.

## Spatial analysis of the risk of occurrence of TL

Spatial GAM models were used to estimate the risks of occurrence of TL in the municipalities in comparison with the average risk in the country [26]. The response variable of the models was the numbers of cases in each year and municipality. The linear predictor comprised a smoothed term from the year of notification, the smoothed term (spline) from the coordinates of the city halls (centroids) of all of the municipalities in the country, and the variable region. A model with a smoothed term of the interaction between notification year × coordinates of municipalities was also evaluated. The offset term was the natural logarithm of the TL-exposed population in each year × coordinates of municipalities. The negative binomial distribution models showed the best fit as described previously. Maps illustrating the time-series of risk of occurrence of TL in the municipalities were constructed, but visualization was limited to locations with positive risk values (> 1). The estimated risk for each municipality was then interpolated to generate a continuous surface. All statistical procedures were performed using R software [27].

## Results

### Incidence and characteristics of TL cases in Brazil, its regions and federative units

The number of new cases of TL reported in Brazil between 2001 and 2020 was 431,885, which corresponds to an overall incidence rate of 226.41 cases/100,000 inhabitants and an average annual incidence rate of 11.32 cases/100,000 inhabitants. A total of 878 deaths from TL were recorded in the target period. The lethality was 0.18%. The North region presented the highest

number of TL cases (182,398) and had the highest incidence rate (1149.73 cases/100,000 inhabitants) but the lowest lethality (0.07%). Within this region, the state of Pará had the highest number of notifications, with more than 70,000 cases, while Acre had the highest incidence rate (2,834.13 cases/100,000 inhabitants) and was the only state in the country to have more than 2,000 cases/100,000 inhabitants (S1 Table).

Along with Acre, the states of Amapá (1,854.30), Roraima (1,733.04), Mato Grosso (1,831.10) and Rondônia (1,539.80) had more than 1,500 cases/100,000 inhabitants. Amazonas (1,056.69) and Pará (933.43) had around 1,000 cases/100,000 inhabitants, while Maranhão (715.43) and Tocantins (704.69) had around 700 cases/100,000 inhabitants. It is noteworthy that Bahia had the third highest number of TL notifications in Brazil (49,180) but with an incidence rate of 350.86 cases/100,000 inhabitants. In the Northeast region, Ceará exhibited the third highest incidence rate of TL (237.66 cases/100,000 inhabitants) after Maranhão and Bahia, while Minas Gerais exhibited the highest incidence rate in the Southeast region (148.53 cases/100,000 inhabitants) (S1 Table).

Tocantins and Mato Grosso presented the highest mortality rates of TL with 4.05 and 2.44 cases/100,000 inhabitants, respectively. Lethality of TL was below 1% for all states with the exception of Sergipe, Rio Grande do Sul and São Paulo, the lethality values of which were 1.37, 1.17 and 1.01%, respectively, despite the low number of TL cases notified in these states (S1 Table).

The demographic, clinical and epidemiological characteristics of TL cases in Brazil, distributed according to region and time period, are detailed S2 Table. The ages of patients were lower in the North and Northeast but, in all regions, the incidence of TL increased with age over the target period, particularly among individuals aged more than 50 years. Although there was a predominance of cutaneous (94.91%) over mucocutaneous (5.50%) forms of the disease in the country as a whole, the proportions of notifications of the mucocutaneous form were slightly higher in the South (11.72%), Southeast (9.52%) and Midwest (7.04%) regions than in the North (5.26%) and Northeast (3.25%) (S2 Table).

The majority of reported cases occurred in males (72.50%), most noticeably in the North (79.25%) and Midwest (80.25%) regions. There was a predominance of non-white individuals among the TL cases in general (65.81%), but with differences between regions that was likely related to the racial mix of the populations. In the North (72.39%) and Northeast (71.86%) regions, the proportion of cases occurring in non-white individuals was greater than 70%, while in the Southeast (47.42%) and Midwest (55.04%) the proportion was around 50%. In the South region, white individuals were predominant among the notified cases of TL (67.43%) (S2 Table).

The proportion of TL cases in rural areas of Brazil (52.64%) was higher in comparison with those in urban settings (43.27%), and similar distributions were observed in the Northeast (64.98%) and North (50.82%) regions. However, in the South (58.83%) and Midwest (56.11%) regions, the urban areas had the highest proportions of notifications, while in the Southeast the numbers of notifications in the urban and rural settings were almost equal. Regarding TL therapy, the proportion of individuals treated with amphotericin B formulations (liposomal or deoxycholate) was highest in the Southeast region (3.80%) and lowest in the Northeast (0.32%), which region also had a significantly lower percentage of cases with laboratory confirmation (64.74%). Data for the other variables were missing from a large proportion of notification forms (S2 Table).

## Temporal evolution of TL cases in Brazil and regions

Age-standardized incidence rates of TL have tended to decline in all Brazilian regions despite fluctuations over the years (Fig 2). Throughout the studied period, the highest incidence rates

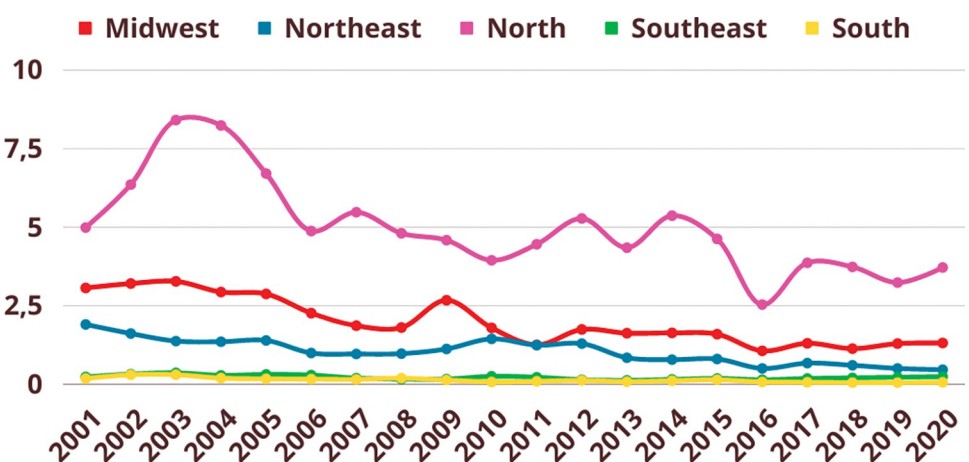

**Fig 2. Age-standardized incidence rates (x 100,000 inhabitants, Y axis) of tegumentary leishmaniasis (TL) in the five Brazilian regions during the period 2001–2020 (X axis).**

occurred in the North region, followed by the Midwest (except for certain periods), Northeast, Southeast and South regions.

GAM (Fig 3) and Joinpoint (Fig 4) regression curves showed a significant linear reduction in the overall incidence rate and risk of occurrence of TL in Brazil during the target period, and the APC and AAPC values confirmed this tendency (Table 1). The lowest incidence rate occurred in 2016, while 2020 showed a higher rate than 2019.

Trends in the incidence rate and risk of occurrence of TL identified in the Joinpoint and GAM regression curves for the whole of Brazil were also observed in the Midwest and South regions (Figs 5 and 6). In the North, the incidence rate could be represented by a straight line with descending slope according to Joinpoint regression (Fig 6), while GAM captured some variations such as an increased risk of occurrence of TL at the beginning of the time-series,

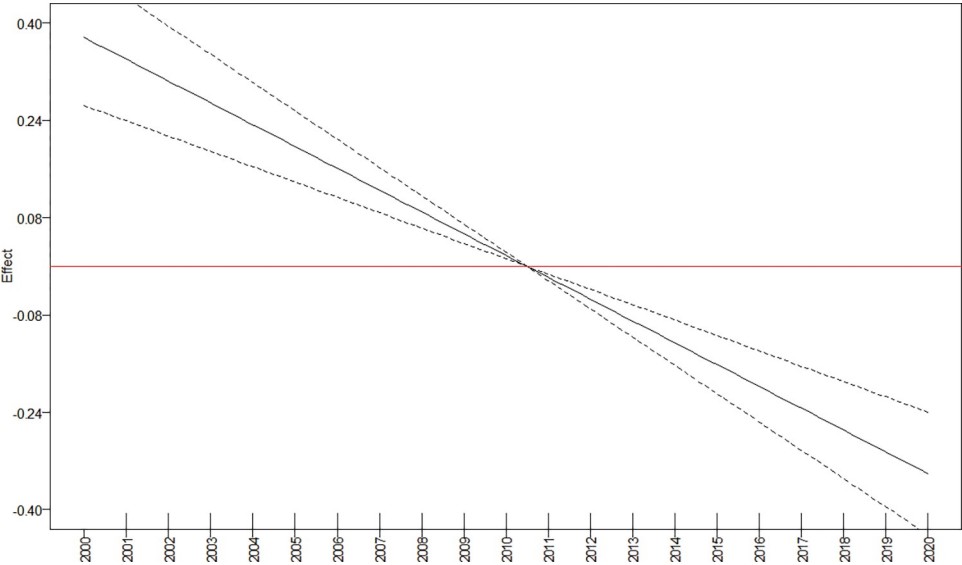

**Fig 3. Risk of tegumentary leishmaniasis in Brazil (Y axis) during the period 2001–2020 (X axis) according to generalized additive models.**

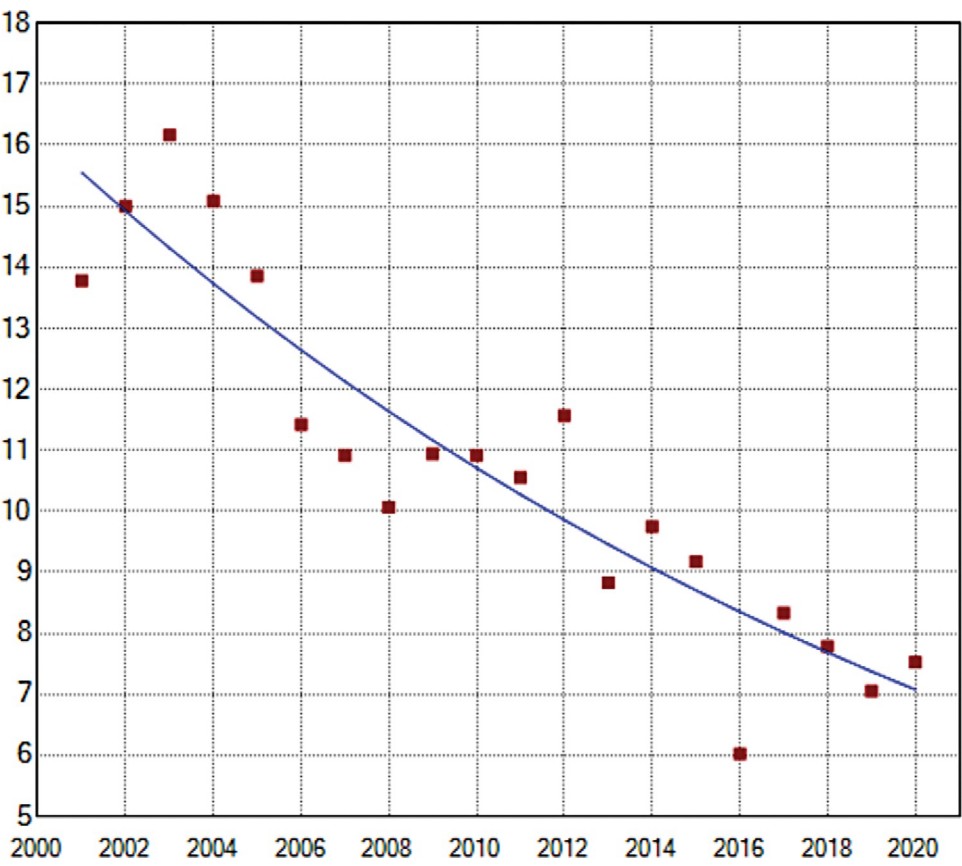

**Fig 4. Incidence rates (x 100,000 inhabitants) of tegumentary leishmaniasis (Y axis) in Brazil during the period 2001–2020 (X axis) according to Joinpoint models.** Points represent annual incidence rates.

with apparent stability between 2010 and 2015 followed by a further sharp reduction (Fig 5). In the Northeast region, incidence rates of TL increased between 2007 and 2010 according to Joinpoint regression (Fig 6), although the risk was statistically significant only in GAM (Fig 5). The Southeast region showed increasing incidence rates from 2014 to 2020 according to Joinpoint (Fig 6), and the risk during this period was statistically significant in GAM. It is

**Table 1. Annual percentage change (APC) and average annual percentage change (AAPC) measures for tegumentary leishmaniasis in Brazil and regions.**

| Country/ region | Time period | APC (95%CIª) | P-value | Trend | AAPC (95%CIª) | Trend |
|---|---|---|---|---|---|---|
| Brazil | 2001–2020 | -4.1 (-4.9 to -3.2) | 0.001 | Decreasing | | |
| Midwest | 2001–2020 | -5.8 (-6.9 to -4.8) | 0.001 | Decreasing | | |
| Northeast | 2001–2007 | -10.4 (-15 to -5.4) | 0.001 | Decreasing | -7.1 (-12.0 to -1.9) | Decreasing |
| | 2007–2010 | +13.5 (-20.3 to +61.8) | 0.451 | Stable | | |
| | 2010–2020 | -10.6 (-13.3 to -7.7) | 0.001 | Decreasing | | |
| North | 2001–2020 | -3.8 (-5.2 to -2.4) | 0.001 | Decreasing | | |
| Southeast | 2001–2014 | -6,5 (-9.4 to -3.6) | 0.001 | Decreasing | -2.4 (-5.8 to 1.2) | Stable |
| | 2014–2020 | +7.3 (-3.1 to +18.9) | 0.162 | Stable | | |
| South | 2001–2020 | -8.2 (-10.1 to -6.2) | 0.001 | Decreasing | | |

ª confidence interval

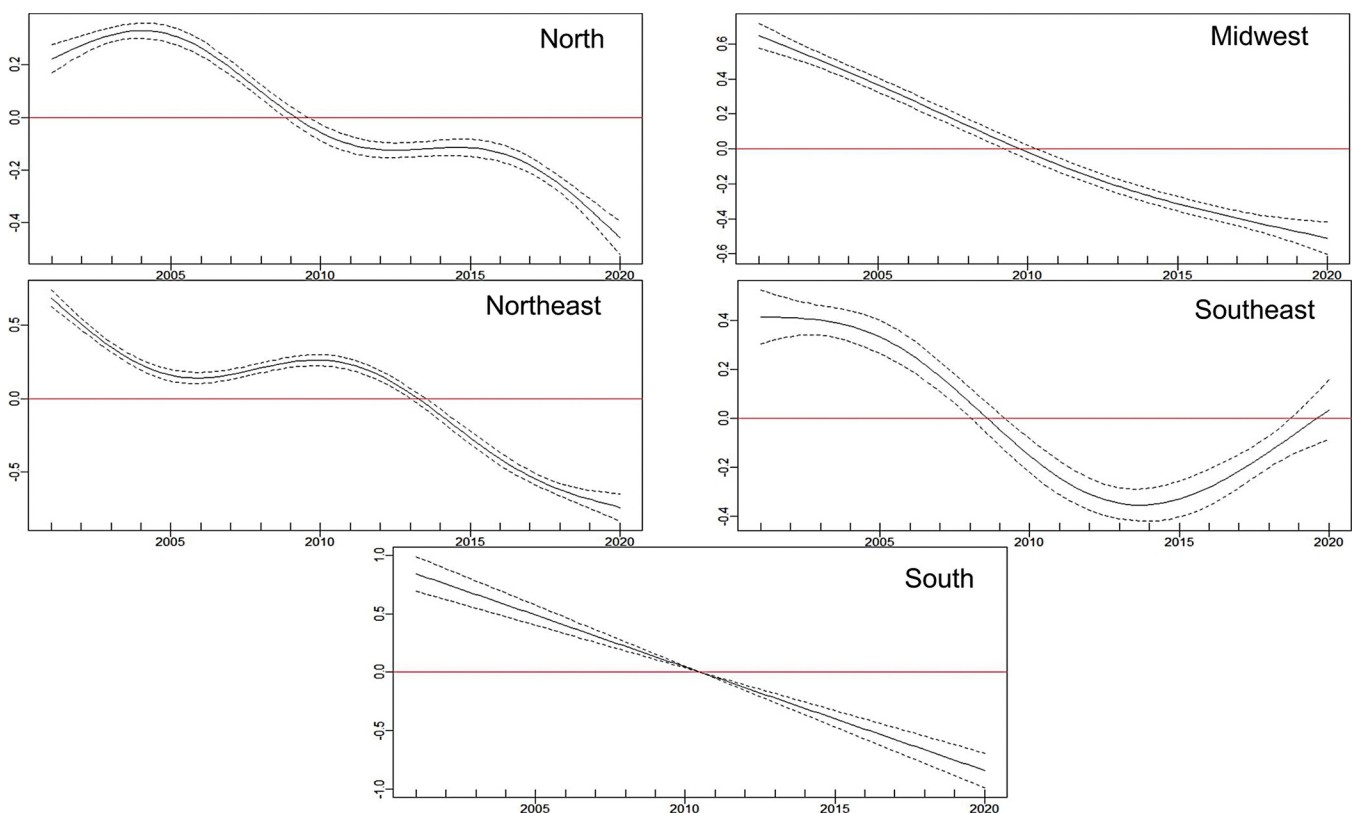

**Fig 5. Risk of tegumentary leishmaniasis (Y axis) in the five Brazilian regions during the period 2001–2020 (X axis) according to generalized additive models.**

noteworthy that, in this region, the incidence rate in 2020 was the highest since 2010 and the annual risk was positive (Fig 5). Although the incidence rates increased during the last years of the time-series, the AAPC value for the Southeast region indicated a stable situation (Table 1).

In general, Joinpoint and GAM regression curves plotted for the federative units reflected the profiles observed in the corresponding regions (S1 Fig). However, some differences could be observed, including: the greater variability in states (Santa Catarina, Paraíba, Rio Grande do Norte and Sergipe) that presented the lowest TL incidence rates and risk of occurrence; the existence of federative units (Federal District, Goiás, Amapá, Rondônia and Tocantins) that exhibited specific periods of increased rates and risks; and states in the Northeast (Alagoas, Ceará, Pernambuco and Piauí) that showed a continuous tendency of reduced incidence rates and risks. In the Southeast, a pattern of increasing incidence rates and risks of occurrence in recent years was observed in Minas Gerais, the state in which cases predominated in the region, but could also be identified in Rio de Janeiro and Espírito Santo. In contrast, São Paulo presented high incidence rates and risks between 2001 and 2003 followed by an inconsistent pattern of decline in subsequent years that was maintained until the end of the time-series.

## Spatial distribution of TL incidence and risks

The decreasing trend of the annual incidence rates in the Brazilian states can be better visualized in choropleth maps (S2 Fig) in which the intensity of the color diminished in most of the federative units over the target period. Acre was the only state presenting incidence rates greater than 80 cases/100,000 inhabitants throughout the time-series. The smoothed spatial

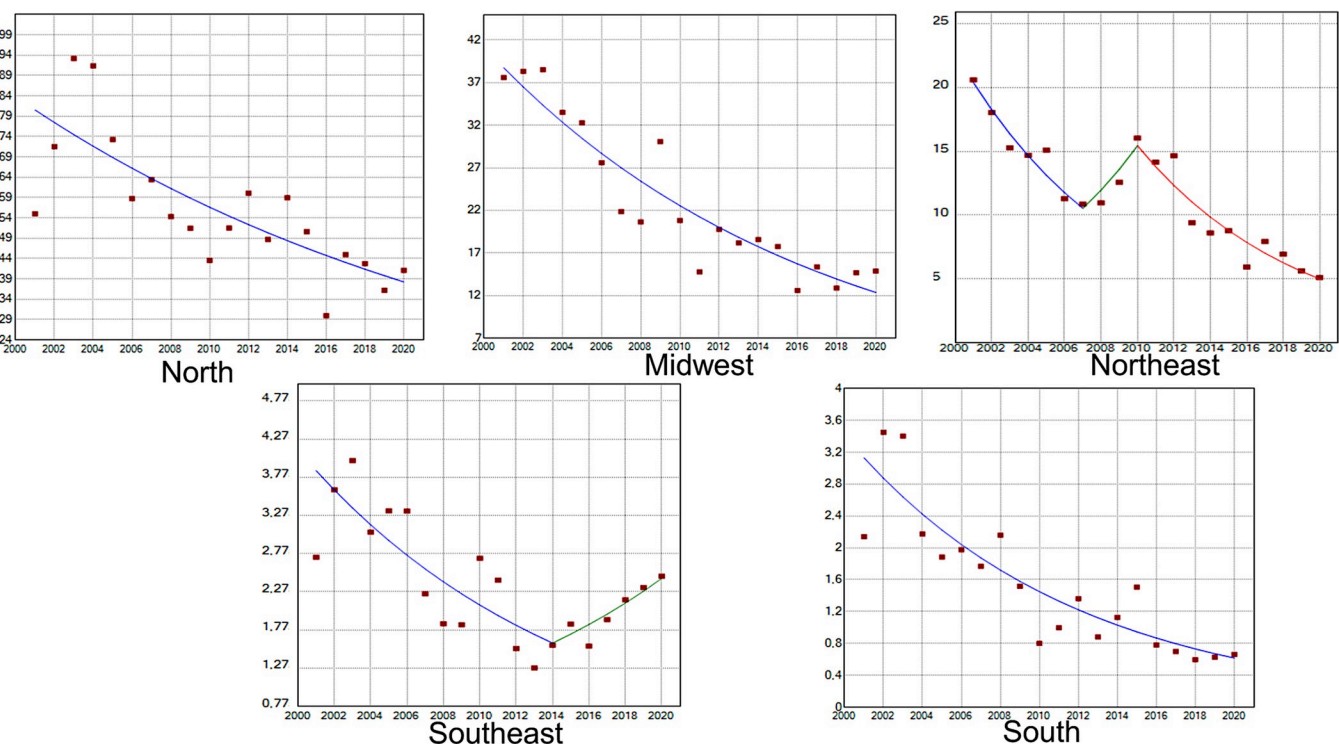

**Fig 6. Age-standardized incidence rates (x 100,000 inhabitants) of tegumentary leishmaniasis (Y axis) in the five Brazilian regions during the period 2001–2020 (X axis) according to Joinpoint models.**

GAM maps (S3 Fig) showed that the greatest risks of occurrence of TL, in relation to the annual averages, throughout the time period were concentrated in a large area covering the North and Midwest regions, as well as some parts of Minas Gerais, Bahia, Maranhão, Piauí and Ceará. Overall, the spatial distribution of risks showed a pattern of stability despite some periods of slight expansion or contraction and minor changes in intensity.

## Discussion

The present study has provided an overall perception of the characteristics of the notified cases of TL in Brazil, its regions and federative units between 2001 and 2020. In general, incidences of the disease have tended to decline in the country, although there have been fluctuations in the rates and risks of occurrence in specific areas. In contrast, the escalating risk in the Southeast region is concerning.

The overall pattern of reduction in the incidence rates of TL identified in Brazil since 2001 is encouraging because it contrasts with the situation in earlier years, more specifically since the 1980s, when TL cases were on the rise [7, 28]. It is pertinent to note that a declining profile in the occurrence of TL has been also observed throughout the American continent, especially in the last 10 years [29]. The downward trend in Brazil may be related to the effectiveness of surveillance and control actions [7], although the effects of these measures have been rarely evaluated in the literature [30–33]. On the other hand, it is possible that the observed trend is due to factors related to the environment, climate, vectors and reservoirs, socioeconomic conditions, individual preventive care or a reduction in the number of susceptible individuals [19, 34–43]. In light of the above, it is important that the dynamics of TL is evaluated regularly and

prospectively, that studies focusing on the determinants of the disease at local and global levels are carried out, and that the effectiveness of control actions is continuously assessed.

Despite the general decline of TL in Brazil, several parts of the country still presented considerable numbers of new cases and/or increased numbers of notifications. It is particularly noteworthy that the lowest number of notifications in the time-series was recorded in 2016, and that the number of cases in 2020 was higher than in 2019 even in the context of the Covid-19 pandemic. Since the clinical course of TL is different from that of Covid-19 [44–46], the effect of the pandemic on the identification and reporting of TL cases may have been smaller than for other diseases [47–49]. On the other hand, changes in the healthcare structure during the pandemic may have compromised the support normally provided to TL patients [29, 45, 50]. Thus, it is possible that severe skin lesions, resulting from cutaneous ulcers that healed without adequate treatment [51], will emerge more frequently in the post-pandemic period. The effects of the pandemic on TL will be better understood as the clinical characteristics of notified cases continue to be evaluated and data regarding the quality of healthcare services provided during period become available.

Despite the overall reduction of TL incidence in all regions of Brazil, a number of localities with increased rates have still been identified, and new cases have emerged in some municipalities that had no previous notifications [52]. From a broader viewpoint, the results presented herein demonstrate that the states of Minas Gerais and, to a lesser extent, Tocantins, Rio de Janeiro and Espírito Santo have seen increased numbers of notifications in recent years. In Minas Gerais, increases have been identified in specific areas presenting the lowest levels of development and significant diversity of sand flies [53, 54], a phenomenon that has been attributed to the migration of TL reservoirs to the vicinity of dwellings [55, 56]. A similar situation has already been reported for Rio de Janeiro [41]. Although some TL reservoirs are more vulnerable to anthropic impact [38], small rodents from forested areas can adapt well to living in close proximity to humans, thereby increasing the risk of disease transmission in regions possessing competent vectors [41].

The presence of TL in all federative units of Brazil affirms its widespread distribution and the importance of disseminating information about the disease to healthcare professionals and residents across the country. However, significant concentrations of cases and high incidence rates of TL are notorious in an area known officially as Brazil's Legal Amazon [57], which encompasses the seven states of the North region together with Mato Grosso (Midwest) and Maranhão (Northeast), and also in the state of Bahia (Northeast). These areas presented the highest risk of occurrence of the disease throughout the target period, and this may be attributed to various factors including the vast extension of forest cover, the wide diversity of TL reservoirs and climatic conditions that are appropriate for the development of vectors, all of which favor the maintenance of the TL transmission cycle [52]. These areas, along with those located in the Southeast region exhibiting an increased trend of the disease, should be prioritized with respect to prevention, control, timely diagnosis and adequate treatment [28, 58]. Furthermore, in order *to* maximize the likelihood of success, it is important to take into account intersectoral interventions considering the particularities of the transmission areas [59], to implement coordinated actions between the different levels of management, to reinforce spatial and temporal analysis in epidemiologic surveillance routines and to engage healthcare professionals and researchers from diverse fields of knowledge [60].

The main characteristics of the studied cases of TL notified in Brazil reproduce the historical pattern, which most commonly involves the cutaneous form of the disease, men of working age and non-white individuals [39, 54, 59]. The differences between the regions with respect to sex and age groups cannot be attributed to population distribution but are more likely associated with different modes of disease transmission [58, 61]. Moreover, the increased incidence

in older age groups, especially in individuals aged over 50 years, may indicate changes in the transmission risk of the individual but might also be a result of the longer life expectancy of the population [62]. Finally, it is noteworthy that a large proportion of the notified cases were reported among urban dwellers even though TL has been linked to forested locations and rural or peri-urban settings with vegetative cover [63]. This result, which likely reflects a lack of standardization in the classification of the areas when completing the notification form [64] or infection events occurring in places other than the residence addresses [65], reinforces the need for further studies on the spatial distribution of TL in Brazil. Research carried out in different Brazilian municipalities has already shown the importance of urban transmission cycles alongside rural and forest cycles [66–70].

Based on the above information, it is recommended that notification forms be updated by including detailed questions concerning the likely place of infection, housing conditions and occupation of the patient. Furthermore, considering the guidelines from the Brazilian Ministry requiring laboratory diagnosis prior to TL therapy [7, 54] and the results of the descriptive analysis reported herein, it is evident that there is a need to improve diagnostic tests in the country, mainly in the Northeast region. Regarding TL therapy, the proportion of individuals who received treatment with amphotericin B was small in all regions, although application of this drug was slightly more frequent in the Southeast and South regions. This difference may be a result of the more advanced ages of infected individuals and the greater severity of the identified cases, but it is of note that these regions also have the highest number of mucocutaneous forms of the disease and the highest lethality rates.

The present study employed secondary data and is, therefore, subject to limitations relating to the quality of the recorded information. The SINAN database consolidates data on notifiable diseases from all over Brazil [52], but the system has a number of deficiencies relating, in particular, to the high proportions of missing data for some variables and differences between regions regarding the completion of the notification forms. In addition, it is possible that the chance of identifying positive cases is lower in regions where TL is uncommon and healthcare professionals are less well acquainted with the disease [4]. Finally, given the diversity in areas and population sizes of Brazilian municipalities and states, the use of aggregated data may limit some of the conclusions obtained and the comparability between the studied areas.

The results reported herein show that TL is in decline in Brazil in general. However, the disease remains extremely relevant to public health because of its widespread occurrence, the existence of states with an upward trend in incidence rate, and the difficulties and inequalities of the affected population to access adequate diagnosis and treatment. The findings of the present study will serve to expand our understanding of the epidemiology of TL in Brazil and are highly relevant to the improvement of public health policies and research project designs.

## Supporting information

**S1 Table. Descriptive measures of new cases of, and deaths from, tegumentary leishmaniasis in Brazil between 2001 and 2020.**
(DOCX)

**S2 Table. Profile and evolution of cases of tegumentary leishmaniasis notified in Brazil and its five major regions between 2001 and 2020.**
(DOCX)

**S1 Fig. Joinpoint analysis (left) and generalized additive models (right) of cases of tegumentary leishmaniasis notified in the 27 federative units of Brazil between 2001 and 2020.**
(DOCX)

**S2 Fig. Choropleth maps of the annual incidence rates (x 100,000 inhabitants) of tegumentary leishmaniasis in the 27 federative units of Brazil between 2001 and 2020.** The maps were built using the free and open source QGIS software (https://www.qgis.org/en/site/) based on shapefiles obtained from Instituto Brasileiro de Geografia e Estatística -IBGE- (https://www.ibge.gov.br/geociencias/organizacao-do-territorio/malhas-territoriais/15774-malhas.html).
(DOCX)

**S3 Fig. Smoothed spatial generalized additive model maps showing areas of greater risk of occurrence of tegumentary leishmaniasis in Brazil between 2001 and 2020 broken down by year.** The maps were built using the free and open source R software (https://www.R-project.org/) based on shapefiles obtained from Instituto Brasileiro de Geografia e Estatística -IBGE- (https://portaldemapas.ibge.gov.br/portal.php#homepage). Black dots represent the centroids of Brazilian municipalities.
(DOCX)

## Author Contributions

**Conceptualization:** Vinícius Silva Belo, Fábio Raphael Pascoti Bruhn, David Soeiro Barbosa, Daniel Cardoso Portela Câmara, Taynãna César Simões, Lia Puppim Buzanovsky, Anna Gabryela Sousa Duarte, Saulo Nascimento de Melo, Diogo Tavares Cardoso, Lucas Edel Donato, Ana Nilce Silveira Maia-Elkhoury, Guilherme Loureiro Werneck.

**Data curation:** Vinícius Silva Belo, Fábio Raphael Pascoti Bruhn, David Soeiro Barbosa, Daniel Cardoso Portela Câmara, Taynãna César Simões, Lia Puppim Buzanovsky, Anna Gabryela Sousa Duarte, Saulo Nascimento de Melo, Diogo Tavares Cardoso, Lucas Edel Donato, Ana Nilce Silveira Maia-Elkhoury, Guilherme Loureiro Werneck.

**Formal analysis:** Vinícius Silva Belo, Fábio Raphael Pascoti Bruhn, David Soeiro Barbosa, Daniel Cardoso Portela Câmara, Taynãna César Simões, Anna Gabryela Sousa Duarte, Saulo Nascimento de Melo, Diogo Tavares Cardoso, Lucas Edel Donato, Guilherme Loureiro Werneck.

**Funding acquisition:** Vinícius Silva Belo, Lucas Edel Donato, Ana Nilce Silveira Maia-Elkhoury.

**Investigation:** Vinícius Silva Belo, Fábio Raphael Pascoti Bruhn, David Soeiro Barbosa, Daniel Cardoso Portela Câmara, Taynãna César Simões, Lia Puppim Buzanovsky, Anna Gabryela Sousa Duarte, Saulo Nascimento de Melo, Diogo Tavares Cardoso, Guilherme Loureiro Werneck.

**Methodology:** Vinícius Silva Belo, Fábio Raphael Pascoti Bruhn, David Soeiro Barbosa, Daniel Cardoso Portela Câmara, Taynãna César Simões, Lia Puppim Buzanovsky, Anna Gabryela Sousa Duarte, Saulo Nascimento de Melo, Diogo Tavares Cardoso, Lucas Edel Donato, Ana Nilce Silveira Maia-Elkhoury, Guilherme Loureiro Werneck.

**Project administration:** Vinícius Silva Belo, Ana Nilce Silveira Maia-Elkhoury, Guilherme Loureiro Werneck.

**Resources:** Vinícius Silva Belo, Fábio Raphael Pascoti Bruhn, David Soeiro Barbosa, Daniel Cardoso Portela Câmara, Taynãna César Simões, Lia Puppim Buzanovsky, Anna Gabryela Sousa Duarte, Saulo Nascimento de Melo, Diogo Tavares Cardoso, Lucas Edel Donato, Ana Nilce Silveira Maia-Elkhoury, Guilherme Loureiro Werneck.

**Software:** Vinícius Silva Belo, Fábio Raphael Pascoti Bruhn, David Soeiro Barbosa, Daniel Cardoso Portela Câmara, Taynãna César Simões, Lia Puppim Buzanovsky, Anna Gabryela Sousa Duarte, Saulo Nascimento de Melo, Diogo Tavares Cardoso, Lucas Edel Donato, Ana Nilce Silveira Maia-Elkhoury, Guilherme Loureiro Werneck.

**Supervision:** Vinícius Silva Belo, Fábio Raphael Pascoti Bruhn, David Soeiro Barbosa, Daniel Cardoso Portela Câmara, Taynãna César Simões, Lia Puppim Buzanovsky, Anna Gabryela Sousa Duarte, Saulo Nascimento de Melo, Diogo Tavares Cardoso, Lucas Edel Donato, Guilherme Loureiro Werneck.

**Validation:** Vinícius Silva Belo, Fábio Raphael Pascoti Bruhn, David Soeiro Barbosa, Daniel Cardoso Portela Câmara, Taynãna César Simões, Lia Puppim Buzanovsky, Anna Gabryela Sousa Duarte, Saulo Nascimento de Melo, Diogo Tavares Cardoso, Lucas Edel Donato, Ana Nilce Silveira Maia-Elkhoury, Guilherme Loureiro Werneck.

**Visualization:** Vinícius Silva Belo, Fábio Raphael Pascoti Bruhn, David Soeiro Barbosa, Daniel Cardoso Portela Câmara, Taynãna César Simões, Lia Puppim Buzanovsky, Anna Gabryela Sousa Duarte, Saulo Nascimento de Melo, Diogo Tavares Cardoso, Lucas Edel Donato, Ana Nilce Silveira Maia-Elkhoury, Guilherme Loureiro Werneck.

**Writing – original draft:** Vinícius Silva Belo, Fábio Raphael Pascoti Bruhn, David Soeiro Barbosa, Daniel Cardoso Portela Câmara, Taynãna César Simões, Lia Puppim Buzanovsky, Anna Gabryela Sousa Duarte, Saulo Nascimento de Melo, Diogo Tavares Cardoso, Lucas Edel Donato, Ana Nilce Silveira Maia-Elkhoury, Guilherme Loureiro Werneck.

**Writing – review & editing:** Vinícius Silva Belo, Fábio Raphael Pascoti Bruhn, David Soeiro Barbosa, Daniel Cardoso Portela Câmara, Taynãna César Simões, Lia Puppim Buzanovsky, Anna Gabryela Sousa Duarte, Saulo Nascimento de Melo, Diogo Tavares Cardoso, Lucas Edel Donato, Ana Nilce Silveira Maia-Elkhoury, Guilherme Loureiro Werneck.

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
