## [Decision Letter · Decision Letter 0]

26 Mar 2023

Dear Dr. Belo,

Thank you very much for submitting your manuscript "Temporal patterns, spatial risks, and characteristics of tegumentary leishmaniasis in Brazil in the first twenty years of the 21st Century" for consideration at PLOS Neglected Tropical Diseases. As with all papers reviewed by the journal, your manuscript was reviewed by members of the editorial board and by several independent reviewers. The reviewers appreciated the attention to an important topic. Based on the reviews, we are likely to accept this manuscript for publication, providing that you modify the manuscript according to the review recommendations. 

Sincerely,

Alberto Novaes Ramos Jr

Academic Editor

Charles Jaffe

Section Editor

Reviewer's Responses to Questions

**Key Review Criteria Required for Acceptance?**

**Methods**

-Are the objectives of the study clearly articulated with a clear testable hypothesis stated?

-Is the study design appropriate to address the stated objectives?

-Is the population clearly described and appropriate for the hypothesis being tested?

-Is the sample size sufficient to ensure adequate power to address the hypothesis being tested?

-Were correct statistical analysis used to support conclusions?

-Are there concerns about ethical or regulatory requirements being met?

Reviewer #1: Page 7, lines 115 to 116: state the year of the population estimate.

Page 8, line 122: in the ethical considerations (lines 112 to 113) the authors inform that the information is free (that is, if open access), however, this link (http://sinan.saude.gov.br/sinan/login/login.jsf) requires login and password.

Page 8, lines 128 to 134: for the calculation of the rates, the authors use the average population for the period, however it was not clear in this one whether the authors used the average of cases in the 20 years, or if they used the sum of cases for the period. As a suggestion, I would use the average of cases or deaths to calculate the rates, since the sum may present relatively high rates for states or administrative regions.

Reviewer #2: The study is relevant and the objective is clearly stated. However, the study design was not declared, although is clear that it consists of an ecological study. The calculation of incidence rates must be clarified. Additionaly, the ranges of incidences rates should follow some pattern form OPAS, WHO or Health Ministry of Brazil.

Reviewer #3: The objective of the study is clear and the study design is appropriate to address the purpose of the study. 

The study analyzed the spatial and temporal patterns of the incidences and risks of occurrence of TL across the five geopolitical regions and 27 federative units in Brazil. 

Cases of TL notified between 2001 and 2020 were obtained from the Information System for Notifiable Diseases of the Brazilian Ministry of Health. Ethical approval for the project was not required because the research involved

data derived from secondary sources that are freely available to the public.

Reviewer #4: This is a very interesting paper, relevant and important, for public health and neglected diseases. 

The results seccion need to clarify why the autors choose to estimates global inciddence, because the recidive is possible in leishmaniosis. The indicator needs the time reference.

The registration of deaths by SINAN is subject to failure, especially since there is no death surveillance. It is more recommended to use data from the Mortality Information System (SIM) for this measure.

**Results**

-Does the analysis presented match the analysis plan?

-Are the results clearly and completely presented?

-Are the figures (Tables, Images) of sufficient quality for clarity?

Reviewer #1: Page 34, fig. 1: include in the map the countries that are neighbours of Brazil.

Pages 35, 36, 37, 38 and 39, fig. 2 and fig. 3: standardize the years of the x axis (Example: in fig. 2 is in the sequence: 2001 to 2020, while in fig.3 we have every 5 years);

Pages 37 and 39, fig. 4 and fig. 6: the data of the figures are already represented in table 1. Evaluate a real need for the figures.

Reviewer #2: The results must be improved. When the results are presented, the absolute and relative frequencies should be highlighted in parentesis. The images in supporting information need be improved either. The incidence rates in coropleth maps should follow the ranges standardized by Health Ministry of Brazil (if exists). There are a plent of maps depicting the incidence at state level year by year. I suggest to represent moving avarages in order to reduce the amount of images. furthermore, the legend of map from description of study area is totally wrong.

Reviewer #3: The analysis used were adequate to support the conclusions of the study. The study described the clinical and epidemiological characteristics of TL cases reported in the country. Joinpoint and generalized additive models were used to establish trends in the evolution of TL during the target period. Clinical and epidemiological characteristics of TL cases were described and analyzed.

Reviewer #4: (No Response)

**Conclusions**

-Are the conclusions supported by the data presented?

-Are the limitations of analysis clearly described?

-Do the authors discuss how these data can be helpful to advance our understanding of the topic under study?

-Is public health relevance addressed?

Reviewer #1: Page 20, lines 372 to 373: in addition to database limitations, the use of data aggregated by states/municipalities may also be a limitation, given the size of municipalities in the North of Brazil, when compared to municipalities in the South/Southeast (for example). The municipalities in the North, besides being municipalities with large territorial areas, have low population indices.

Reviewer #2: I suggest that authors highlight the implications of findings for epidemiologic surveillance. It was mentioned that the results could contribute to improve therapeutic actions. However, I would rather to focus on the intersectoral strategies to reduce the iniquities among the regions and the importante of adoption of spatial and temporal tools in surveillance.

Reviewer #3: The conclusions are supported by the presented data and analysis performed. Study limitations are clearly described. The main limitation refers to the quality of the information recorded in the secondary data analyzed in the study. According to the authors the findings of the study will serve to expand the understanding of the epidemiology of TL in Brazil and are highly relevant to the improvement of public health policies. Despite the general decline of TL in Brazil, several parts of the country still presented considerable numbers of cases and/or increased numbers of notifications.

Reviewer #4: (No Response)

**Editorial and Data Presentation Modifications?**

Reviewer #1: (No Response)

Reviewer #2: Minor revision.

Reviewer #3: Line 77: …”significant public health problem and requires compulsory notification” 

Please, add a reference on compulsory notification diseases in Brazil. The reference 6 (World Health Organization. Leishmaniasis) is about general LT information.

Lines 101-103: The reference 19; Bruhn et al., is about visceral leishmaniasis. I am not sure if this reference is in accordance with the cited text.

Please, change the name of Central-west to Midwest region in Figure 2. All the others figures and tables the name used of the region is Midwest.

Reviewer #4: (No Response)

**Summary and General Comments**

Reviewer #1: Page 3, line 36: I suggest that the authors briefly inform in the method how the global incidence rate was calculated - did they use the average of cases in the period or the sum of cases?

Reviewer #2: The findings of sutdy are important to provide a overall scenario of TL in the last two decades. The authors could have performed cluster analysis to demonstrate the spatial correlation, but I think that the analysis carried out are sufficent to achieve the main objetive.

Reviewer #3: The study analyzed 431,885 cases of TL reported in Brazil between 2001 and 2020. The manuscript is well-written and presents interesting results on clinical and epidemiological characteristics of TL cases and spatial and temporal patterns of TL. The discussion placed this study in context and provides a thorough review of the results found in the spatial and temporal analysis and clinical and epidemiological characteristics of TL cases. Considering the relative scarcity of good quality studies on TL, I believe this article will provide an important contribution to the literature.

Reviewer #4: (No Response)

PLOS authors have the option to publish the peer review history of their article (what does this mean?). If published, this will include your full peer review and any attached files.

Reviewer #1: No

Reviewer #2: No

Reviewer #3: No

Reviewer #4: No

Figure Files:

Data Requirements:

Reproducibility:

References

---

## [Decision Letter · Decision Letter 1]

22 May 2023

Dear Dr. Belo,

We are pleased to inform you that your manuscript 'Temporal patterns, spatial risks, and characteristics of tegumentary leishmaniasis in Brazil in the first twenty years of the 21st Century' has been provisionally accepted for publication in PLOS Neglected Tropical Diseases.

Best regards,

Alberto Novaes Ramos Jr

Academic Editor

Charles Jaffe

Section Editor

Reviewer's Responses to Questions

**Key Review Criteria Required for Acceptance?**

**Methods**

-Are the objectives of the study clearly articulated with a clear testable hypothesis stated?

-Is the study design appropriate to address the stated objectives?

-Is the population clearly described and appropriate for the hypothesis being tested?

-Is the sample size sufficient to ensure adequate power to address the hypothesis being tested?

-Were correct statistical analysis used to support conclusions?

-Are there concerns about ethical or regulatory requirements being met?

Reviewer #1: (No Response)

Reviewer #2: The study is innovative and well written. I suggest that the description of the study design be added in the method section and that the reason for not using municipalities as the study's analysis units be explained in the study's limitations.

Reviewer #3: The objective of the study is clear and the study design is appropriate to address the purpose of the study.

**Results**

-Does the analysis presented match the analysis plan?

-Are the results clearly and completely presented?

-Are the figures (Tables, Images) of sufficient quality for clarity?

Reviewer #1: (No Response)

Reviewer #2: (No Response)

Reviewer #3: The analyzes were adequate to support the conclusions of the study. The study described the clinical and epidemiological characteristics of TL cases reported in the country.

**Conclusions**

-Are the conclusions supported by the data presented?

-Are the limitations of analysis clearly described?

-Do the authors discuss how these data can be helpful to advance our understanding of the topic under study?

-Is public health relevance addressed?

Reviewer #1: (No Response)

Reviewer #2: I recommend that the authors expose in the discussion the contribution that these results can provide to the achievement of the sustainable development goals.

Reviewer #3: The conclusions are supported by the presented data and analysis performed.

**Editorial and Data Presentation Modifications?**

Reviewer #1: (No Response)

Reviewer #2: (No Response)

Reviewer #3: "Accept".

**Summary and General Comments**

Reviewer #1: (No Response)

Reviewer #2: (No Response)

Reviewer #3: The manuscript is well-written and presents interesting results on clinical and epidemiological characteristics of TL cases and spatial and temporal patterns of TL.

PLOS authors have the option to publish the peer review history of their article (what does this mean?). If published, this will include your full peer review and any attached files.

Reviewer #1: **Yes: **Anderson Fuentes Ferreira

Reviewer #2: No

Reviewer #3: No

---

## [Editor Report · Acceptance letter]

2 Jun 2023

Dear Dr. Belo,

We are delighted to inform you that your manuscript, "Temporal patterns, spatial risks, and characteristics of tegumentary leishmaniasis in Brazil in the first twenty years of the 21st Century," has been formally accepted for publication in PLOS Neglected Tropical Diseases.

Best regards,

Shaden Kamhawi

co-Editor-in-Chief

Paul Brindley

co-Editor-in-Chief
